# Multi-Wavelength Photobiomodulation Ameliorates Sodium Iodate-Induced Age-Related Macular Degeneration in Rats

**DOI:** 10.3390/ijms242417394

**Published:** 2023-12-12

**Authors:** Hyeyoon Goo, Min Young Lee, Yea-Jin Lee, Sangkeun Lee, Jin-Chul Ahn, Namgue Hong

**Affiliations:** 1Department of Medical Laser, Graduate School of Medicine, Dankook University, Cheonan 31116, Republic of Korea; ghy1204@hanmail.net; 2Beckman Laser Institute-Korea, Dankook University, Cheonan 31116, Republic of Korea; eyeglass210@gmail.com (M.Y.L.); yjlee0109@dankook.ac.kr (Y.-J.L.); 3Department of Otolaryngology-Head & Neck Surgery, College of Medicine, Dankook University Hospital, Dankook University, Cheonan 31116, Republic of Korea; 4MEDI-IOT Co., Ltd., Seoul 02708, Republic of Korea; sklee.mediiot@gmail.com

**Keywords:** age-related macular degeneration, photobiomodulation, multi-wavelength, oxidative stress, sodium iodate

## Abstract

Age-related macular degeneration (AMD) is a global health challenge. AMD causes visual impairment and blindness, particularly in older individuals. This multifaceted disease progresses through various stages, from asymptomatic dry to advanced wet AMD, driven by various factors including inflammation and oxidative stress. Current treatments are effective mainly for wet AMD; the therapeutic options for dry AMD are limited. Photobiomodulation (PBM) using low-energy light in the red-to-near-infrared range is a promising treatment for retinal diseases. This study investigated the effects of multi-wavelength PBM (680, 780, and 830 nm) on sodium iodate-induced oxidatively damaged retinal tissue. In an in vivo rat model of AMD induced by sodium iodate, multi-wavelength PBM effectively protected the retinal layers, reduced retinal apoptosis, and prevented rod bipolar cell depletion. Furthermore, PBM inhibited photoreceptor degeneration and reduced retinal pigment epithelium toxicity. These results suggest that multi-wavelength PBM may be a useful therapeutic strategy for AMD, mitigating oxidative stress, preserving retinal integrity, and preventing apoptosis.

## 1. Introduction

Age-related macular degeneration (AMD) is a widespread and devastating eye condition that poses a significant global health challenge. AMD is the principal cause of visual impairment and blindness, particularly in older individuals [1]. AMD progression is characterized by distinct stages, starting with (often asymptomatic) dry AMD and then advancing to late stages associated with severe vision loss attributable to the disruption of the retinal anatomy and eventual photoreceptor degeneration [2,3]. In the late stages, notably in wet AMD, invasion of new choroidal blood vessels into the subretinal space causes edema and rapid visual deterioration [4].

AMD has several risk factors such as older age, cigarette smoking, previous cataract surgery, and familial history of AMD [5]. Currently, 52 gene variants have been found in relation to AMD [6], but no clinical gene therapy is currently available. Some of these risk factors are modifiable and, to prevent AMD, stopping cigarette smoking and losing body weight would be essential. But other risk factors are not modifiable. A prophylactic approach using an antioxidant has been proposed for dry AMD but further studies suggest that evidence is still inconclusive [7]. A deeper understanding of the multifaceted pathogenesis of AMD, which involves metabolic disorders, immunological changes, inflammation, and oxidative stress, is essential to enhancing our ability to manage and mitigate this debilitating condition. A deep understanding of the intricate cellular pathways active in both dry and wet AMD is a prerequisite. Current treatments are effective mainly for wet AMD; dry AMD remains a formidable challenge, and there are few therapeutic options [8]. There has been significant advancement in therapeutic options for wet AMD. Starting from the introduction of photodynamic therapy (photosensitizer and laser) several decades earlier, which is not used alone due to the high recurrence rate and vision loss, advanced anti-VEGF agent development has been followed and it is currently used for wet AMD patients in combination with photodynamic therapy [7]. However, there is currently no therapy for dry AMD. Several approaches using either a human monoclonal antibody, mitochondrial protective compound, or rapamycin are under clinical trials and expected to slow the progression of the disease [7]; however, no cure is expected currently. Development of therapy is burdened due to a complex interplay of risk factors including older age, oxidative stress, genetic characteristics, and chronic inflammation. These factors must be taken into account when developing targeted interventions alleviating the burden of a vision-threatening disease [9,10,11]. Excessive reactive oxygen species (ROS) production compromises the antioxidant defenses and metabolic functions of retinal pigment epithelium (RPE) cells, triggering lysosomal degradation that culminates in apoptosis and AMD progression [12,13]. As the population ages, efforts to understand the intricate mechanisms of AMD are essential to developing innovative, minimally toxic, therapeutic agents that improve visual outcomes.

Photobiomodulation (PBM), also termed low-level laser therapy, has emerged as a promising approach for AMD alleviation [14]. PBM uses low-energy light, typically in the red-to-near-infrared range (600–1000 nm), to exert beneficial effects in target cells or tissues [15,16]. Although AMD treatment remains challenging, recent studies have highlighted the therapeutic utility of PBM in patients with various retinal diseases, including dry AMD [14,17,18,19]. PBM, using low-energy red light, improved the visual acuity and contrast sensitivity, and reduced drusen volume, and possibly attenuated the progression of dry AMD and also enhanced visual function [20]. In animal models of diabetic macular edema (DME), PBM reduced foveal thickness, ameliorated retinal microvascular occlusion, and suppressed neovascularization [21,22].

PBM is believed to primarily target cytochrome C oxidase (CCO), a key protein in the mitochondrial respiratory chain, thus enhancing energy metabolism and ATP production [23]. Nevertheless, the means by which PBM affects cellular and molecular mechanisms remains unclear. The changes in the parameters of PBM may affect the target effects, especially different wavelengths of PBM that can induce alternative outcomes [24]. In terms of the ophthalmologic application of PBM, some wavelengths of PBM have been associated with reductions in outer retinal inflammation and oxidative stress, which are known risk factors for AMD development and progression [25]. Most studies used red light (670 nm). The light of other wavelengths (such as 780 nm) has been also shown therapeutic effect on wet AMD [26]. The combination and simultaneous application of multiple wavelengths of light could lead to favorable outcomes due to the lower energy use and the lower chance of complication from over exposure to a single wavelength of light.

Our primary objective was to investigate the effects of multi-wavelength PBM (680, 780, and 830 nm) on oxidatively damaged retinal tissue. Although several studies have explored the effects of single-wavelength PBM, few have evaluated multi-wavelength PBM, especially in terms of its effects on retinal oxidative stress. The effects of different wavelengths may be distinct. Therefore, we evaluated how multi-wavelength PBM affects retinal health in the presence of oxidative stress, and how it may offer valuable insights into potential therapeutic and preventative applications of PBM in patients with retinal disorders.

## 2. Results

### 2.1. Multi-Wavelength PBM Does Not Damage RPE Cells In Vitro

The safety of multi-wavelength PBM was initially assessed using retinal pigment epithelial cells (ARPE-19 line) in vitro. Figure 1a shows the cell viability 24 h after exposure to multi-wavelength PBM with energies of 0, 1.125, 2.25, 4.5, 9.0, and 18 J/cm^2^. Cell viability was unaffected; multi-wavelength PBM did not induce significant cytotoxicity. We then measured cell viability at 24, 48, and 72 h after multi-wavelength PBM (Figure 1b) and found that it was similarly unchanged. Furthermore, we investigated the cytotoxicity of each individual wavelength used in multi-wavelength PBM on RPE cells (Appendix A). No cytotoxicity was observed for each of the individual wavelengths.

### 2.2. In Vivo Rat Model of Sodium Iodate-Induced AMD

We established a rat model with NaIO_3_-induced retinal damage. From 1 to 7 days after NaIO_3_ treatment, histological analyses (hematoxylin and eosin [H&E] staining) were used to determine the thickness of the entire retina and the outer nerve layer (ONL). Representative images of retinal sections 7 days after NaIO_3_ treatment are shown in Figure 2a. The total retinal thickness after 4 and 7 days of NaIO_3_ treatment was significantly lower compared with sham-treated rats (*p* < 0.05, Figure 2b). The ONL thickness after 3, 4, and 7 days of treatment was significantly lower compared with sham-treated rats (*p* < 0.0001, Figure 2b). The inner photoreceptor (IS) and outer photoreceptor (OS) segments were disrupted after 7 days of treatment.

### 2.3. Multi-Wavelength PBM Protects against Retinal Degeneration

Multi-wavelength PBM protected the retinal layers and partially mitigated IS/OS disruption (Figure 3a). Retinal degeneration was quantified by measuring ONL thickness; this was reduced in NaIO_3_-treated rats (*p* < 0.05), but the decrement was significantly reduced by multi-wavelength PBM (*p* < 0.001, Figure 3b). The thicknesses of the ganglion cell layer (GCL) and nerve fiber layer (NFL), which receive visual signals and then transmit them to the brain, were also measured. Multi-wavelength PBM prevented decreases in ONL and GCL+NFL thickness (*p* < 0.05, Figure 3c). In summary, multi-wavelength PBM prevented NaIO_3_-induced retinal degeneration.

### 2.4. Multi-Wavelength PBM Reduces NaIO_3_-Induced Retinal Apoptosis

Previous studies found that NaIO_3_-induced retinal damage by triggering apoptosis [27]. Thus, we then assessed retinal apoptotic status in NaIO_3_-exposed rats. Figure 4a shows many TUNEL-positive cells (green) (primarily) in the retinal ONLs of NaIO_3_-exposed rats. Multi-wavelength PBM significantly reduced the cell numbers (Figure 4b; *p* < 0.001). Thus, multi-wavelength PBM inhibited NaIO_3_-induced retinal apoptosis.

### 2.5. Multi-Wavelength PBM Prevents Depletion of Rod Bipolar Cells in the Inner Nerve Layer

Progressive degeneration of retinal ganglion cells is common during the development of visual disorders. Damage to rod bipolar cells precedes that to retinal ganglion cells as the retina deteriorates [28]. Therefore, we immunohistochemically stained for PKCα, a marker of rod bipolar cells in the retinal inner nerve layer (INL) (Figure 5a). NaIO_3_ significantly reduced the number of bipolar cells (*p* < 0.0001, Figure 5b). However, multi-wavelength PBM significantly increased the number of cells (*p* < 0.05, Figure 5b). Multi-wavelength PBM thus inhibited INL rod bipolar cell depletion.

### 2.6. Multi-Wavelength PBM Inhibits Photoreceptor Degeneration and Reduces RPE Toxicity

We used immunofluorescence staining to quantitate the levels of rhodopsin, a marker of photoreceptor cells, and RPE65, a marker of the RPE layer. NaIO_3_ significantly reduced the fluorescence intensity of rhodopsin (*p* < 0.0001) and RPE65 (*p* < 0.01). Multi-wavelength PBM significantly increased the fluorescence intensity in both cases (*p* < 0.01 and *p* < 0.05, respectively). Thus, multi-wavelength PBM-mediated reduction in the oxidative stress inhibited NaIO_3_-induced photoreceptor cell degeneration, which in turn reduced the RPE toxicity.

## 3. Discussion

Global aging is associated with increased prevalence of age-related diseases, particularly AMD, which significantly reduces quality of life. There is no cure for AMD, and effective anti-AMD agents are urgently required. Recent research focused on PBM as a novel treatment for retinal degeneration. PBM exerts anti-inflammatory and antioxidant effects and has been widely used to treat various diseased tissues [29,30,31,32]. Recent research has employed PBM to reduce RPE degeneration, with promising results [26,33]. We first assessed whether multi-wavelength PBM was toxic to retinal epithelial cells (Figure 1) and found that this was not the case. Our low-power multi-wavelength PBM protocol did not damage eyes. We then used NaIO_3_ to induce retinal degeneration in rats; NaIO_3_ severely damaged the retinal ONL and IS/OS junction (Figure 2) [34,35]. NaIO_3_ triggers rapid-onset RPE damage, particularly at high doses (50–100 mg/kg). Even at lower doses, NaIO_3_ disrupted rodent retinal function by triggering retinal degeneration [36,37]. We found that intravenous NaIO_3_ at 35 mg/kg caused retinal degeneration within 7 days. Importantly, multi-wavelength PBM effectively ameliorated NaIO_3_-induced retinal degeneration in vivo.

In the current study, we have used normal aged rats as a model for AMD. Considering the fact that AMD is a pathology related to the aging process, it might be necessary to use older aged animals to induce disease for an animal model. However, in that case, there is the difficulty in categorizing the control group. The best option would be to use some old animals without the application of NaIO_3_. But in this aged control animal, an age-related change in the macular could exist even without any pharmacologic agents. As such, many studies, along with our own, used young age rats to induce AMD in animal models [38,39]. The experimental design of 7 days also requires additional discussion. The presence of drusen within the macula, which is the hallmark of AMD [7], was only observed until 7 days in the present experiment. Therefore, a longer experiment was not possible due to this reason.

The irradiation of PBM has been studied widely and many studies have shown that it provides safety for a variety of different cells. One study from our group has shown that its effect on the HEI-OC1 cell line, which is a sensory cell line, persists only for 24 h and does not have a prolonged effect [40]. In the current study, we have monitored the cells more than 3 days and there was no toxicity; instead, an increase in cell viability was observed. There are several clinical studies that show the safety of this method in humans. In a review for PBM for head and neck cancer therapy in clinics, no complication development was observed until 18 months in all the studies using similar parameters [41]. These results indicate that performing PBM irradiation on a patient would be safe; however, prior to actual clinical application, a safety assessment using more specific conditions and better animal models is necessary.

AMD is associated with several risk factors, many of which are closely linked to increased levels of reactive oxygen species (ROS). The literature emphasizes the role played by excessive ROS in AMD development, which culminates in RPE cell apoptosis [42]. In animal AMD models, NaIO_3_-induced AKT activation in RPE cells was partly dependent on ROS production, ultimately causing cell death [43,44,45]. To explore the effects of multi-wavelength PBM on the retina, we established an in vivo model of AMD induced by NaIO_3_. It is known that PBM of various wavelengths reduces ROS levels and thus cellular oxidative stress [24,46]. In our in vitro experiment, although we observed that multi-wavelength PBM tends to decrease ROS changes in RPE cells after oxidative stress-induced by hydrogen peroxide, we did not obtain significant results (Appendix A). However, we observed that oxidative stress induced by NaIO_3_ triggered apoptosis of the GCL and ONL. However, multi-wavelength PBM effectively prevented this (Figure 3 and Figure 4). Our work, together with previous findings, suggests that the antioxidant effect of PBM effectively mitigates oxidative stress, such as the increase in ROS levels triggered by NaIO_3_, thereby preventing apoptosis of the ONL and GCL.

Previous studies found that rod bipolar cells were damaged prior to GCL injury as the retina degenerated [28]. We focused on rod bipolar cells of the INL; we used the PKCa marker of such cells to assess whether PBM increased GCL thickness (Figure 5). We found that PBM effectively protected INL PKCa-expressing cells from NaIO_3_-induced toxicity, thus protecting the GCL.

NaIO_3_-induced RPE degeneration has been associated with the production of pigment-rich substances, and reductions in the thicknesses of both the OS and IS segments and the ONL [36,47,48]. Exposure of RPE cells to NaIO_3_ disrupted cellular integrity [49]. We found that the rhodopsin and RPE65 fluorescence intensity in NaIO_3_-treated rats was significantly reduced by day 7. However, multi-wavelength PBM effectively prevented such retinal degeneration (Figure 6), suggesting that PBM inhibited the effects of oxidative stress on RPE cells and photoreceptors, which in turn enhanced ONL photoreceptor survival.

Multi-wavelength PBM protocols may be better than single-wavelength methods. The former protocols treat tissues at different depths within the retina. Light of 680 nm enhances the proliferation and activity of rod bipolar and ganglion cells; light of 780 and 830 nm effectively reduces oxidative stress in deeper RPE cells and photoreceptors. Further studies using both cellular and animal models of retinal degeneration are required to confirm the feasibility and efficacy of our multi-wavelength approach. Given our results and the future prospects, we suggest that multi-wavelength PBM could reduce the severity of several AMD pathologies and that it is a promising therapeutic strategy for patients exhibiting retinal degeneration.

There is an ongoing clinical study using multi-wavelength light therapy for AMD (Trial: Clinicaltrial.Gov Registration Identifier: NCT03878420). In the clinical study of 44 patients, the safety of PBM was confirmed and slow disease progression was observed [50]. Despite this favorable outcome, the use of subclinical animal studies to support this result is insufficient. In the document regarding this clinical analysis, authors have claimed that there is sufficient evidence of multi-pathway improvement of inflammatory control using PBM to apply this to AMD since there is substantial evidence of the role of inflammation in its pathogenesis [51]. Even though there are many studies that have proven favorable outcomes with PBM in other ophthalmologic diseases [52,53,54,55], very few studies have used AMD in animal models. In addition, these studies used single wavelength PBM [56,57] to show its efficiency. Thus, along with the clinical study, the present animal study would be necessary to support the usage of PBM for such a disease without a cure treatment modality.

There are several limitations of this study. It could be the lack of a detailed in vitro investigation, since the pathomechanism of PBM is not currently clear, and it would have been better to include a complex cellular experiment to describe the mechanism in detail. Another limitation would be the mismatch between the animal model and the clinical setting because, as has been noted in the discussion, the pathology of AMD is almost certainly age-related, and thus using the older animals would be more suitable for an experiment.

As has been noted earlier, this is the only animal study using the multiple wavelength PBM to show its beneficial effect on dry AMD. Nevertheless, there are two well-established clinical studies that show its safety and one well-established study that shows its efficiency in reducing the progression of AMD. This study further provides evidence that the treatment is effective for AMD, especially by providing histologic outcomes, which has thus far not been available in clinical experimental settings.

## 4. Materials and Methods

### 4.1. Cell Culture

Retinal epithelial cells of the ARPE-19 line were cultured in Dulbecco’s modified Eagle’s Medium (DMEM; Gibco, Grand Island, NY, USA) with 1% (*w*/*v*) penicillin and streptomycin (Gibco), and 5% (*v*/*v*) heat-inactivated fetal bovine serum (FBS; Equitech-Bio Inc., Kerrville, TX, USA) at 37 °C under 5% (*v*/*v*) CO_2_ in a humidified chamber. The medium was changed every 2–3 days. Cells were sub-cultured at a confluence of 70–80% after detachment with 0.25% (*w*/*v*) trypsin-EDTA (Corning, Manassas, VA, USA).

### 4.2. Multi-Wavelength PBM

Multi-wavelength PBM used light of wavelengths 680, 780, and 830 nm (Figure 7a). The light source had a 3.5 × 3.5 mm ceramic package and a 1000 × 1000-µm AlGaAs LED chip. The LED spectrum was measured using a spectrometer (USB4000; Ocean Optics, Orlando, FL, USA) (Figure 7b). The LED power was assessed by placing a laser fiber spectroradiometer with a diffuser in the multi-wavelength PBM irradiation port.

### 4.3. Cell Viability and Proliferation

Cell viability and proliferation were determined using the MTT assay (Figure 7c). ARPE-19 cells were seeded into 12-well culture plates, incubated overnight, and subjected to multi-wavelength PBM at intensities of 1.125, 2.25, 4.5, 9, and 18 J/cm^2^ at a distance of 17 mm. To assess proliferation, after irradiation with multiple LEDs at 4.5 J/cm^2^, cells were cultured under 5% (*v*/*v*) CO_2_ at 37 °C for 24, 48, and 72 h. Then, 100 μL of 2,5-diphenyl-2H-tetrazolium bromide (MTT) solution (5 mg/mL) was added to each well. After 4 h, the medium was removed and 100 μL dimethyl sulfoxide (DMSO) was added to dissolve the purple crystals. Cell viability and proliferation were determined by measuring absorbance at 570 nm using an ELISA reader (Synergy HTX; BioTek, Santa Clara, CA, USA).

### 4.4. In Vivo Experimental Schedule

All animal protocols were approved by the Institutional Animal Care and Use Committee of the Dankook University Medical School (approval no. DKU 23-020). Six-week-old male Sprague–Dawley rats were obtained from Nara Biotech (Pyeongtaek, Republic of Korea) and housed under a 12 h light/12 h dark cycle at 23 ± 1 °C and 44 ± 2% relative humidity. The rats were divided into three groups (control, dry AMD, and dry AMD+LED groups). A schematic of the in vivo study is shown in Figure 7d. The control group was not treated, while the dry AMD and dry AMD+LED groups received 35 mg/kg NaIO_3_ injections into the tail vein. To establish dry AMD, NaIO_3_ injections (35 mg/kg, i.p.) were performed. The eyeballs were removed 1, 2, 3, 4, and 7 days after for analysis. In the dry AMD group, eyes were irradiated five times for 2 min each time with the LED (4.5 J/cm^2^), and, 7 days after injection, the eyeballs removed and the tissues were stained.

### 4.5. Histology

Rats were sacrificed via CO2 asphyxiation at the end of the experiment and the eyeballs were excised, fixed in 4% (*v*/*v*) paraformaldehyde overnight at 4 °C, dehydrated by incubation in a series of sucrose solutions (10, 20, and 30% *w*/*v*) overnight at 4 °C, and embedded in OCT compound (Sakura, Torrance, CA, USA). Slices (4 μm thick) were prepared using a cryostat (CM1850; Leica, Wetzlar, Germany) and mounted on silane-coated slides. The sections were deparaffinized in xylene, rehydrated through a series of graded alcohol baths, and stained with H&E. Retinal thickness was measured under a microscope (BX53; Olympus, Tokyo, Japan) running CellSence software (2.3 ver.; Olympus, Tokyo, Japan) and quantified with Image J software (1.52n ver.; National Institutes of Health, Bethesda, MD, USA).

### 4.6. TUNEL Assay

Tissue slides were washed twice with PBS and permeabilized with Triton X-100 (Sigma-Aldrich, St. Louis, MO, USA) for 5 min at room temperature. After washing with PBS at room temperature, slides were covered with 100 μL 1 × reaction buffer, incubated at room temperature for 5–10 min, and washed with PBS. Fifty microliters of staining solution prepared according to the manufacturer’s instructions (5 μL of dUTP-conjugated dye, 10 μL of 5 × reaction buffer, 1 μL of TdT recombinant enzyme, and 35 μL of deionized water (BioActs, Incheon, Republic of Korea) was added to each slide followed by incubation at 37 °C for 1 h. The slides were rinsed three times with PBS at room temperature for 5 min to remove unreacted dye-dUTP, incubated with propidium iodide (PI) solution (1 μg/mL) for 15 min at room temperature in the dark, and washed with PBS. VectaShield mounting solution (Vector Laboratories Inc., Newark, CA, USA) was then added. All sections were examined under a confocal microscope, and stained cells were counted.

### 4.7. Immunohistochemistry

Frozen sections were washed twice with 0.1 M phosphate-buffered saline (PBS) for 5 min each time, blocked with 5% (*w*/*v*) bovine serum albumin (BSA) in PBS for 1 h, incubated with anti-rhodopsin (AB5417; Abcam, Cambridge, UK), anti-RPE65 (ab231782; Abcam, Cambridge, UK), and anti-PKCα (SC-10800; Santa-Cruz, Santa Cruz, CA, USA) primary antibodies overnight at 4 °C, and then incubated with Alexa Fluor 488-conjugated anti-rabbit IgG (A11008; ThermoFisher, Waltham, MA, USA) or Alexa Fluor 555-conjugated anti-mouse IgG (A21422; ThermoFisher, Waltham, MA, USA) for 1 h at room temperature. Immunostained tissues were mounted with VectaShield Antifade mounting medium containing DAPI (H-1200; Vector Laboratories Inc., Newark, CA, USA). Alexa Fluor 488 (excitation wavelength, 488 nm; emission wavelength, 520 nm) and Alexa Fluor 555 (excitation, 561 nm; emission, 568 nm) signals were imaged using a confocal microscope (FV3000; Olympus).

### 4.8. Statistics

Data are reported as mean ± standard errors of the mean (SEM) All analyses were performed using GraphPad Prism software (ver. 10.0.2; GraphPad Software Inc., La Jolla, CA, USA). Data were subjected to one-way analysis of variance (ANOVA) followed by the Bonferroni post hoc test. A *p*-value < 0.05 was considered statistically significant.

## Figures and Tables

**Figure 1 ijms-24-17394-f001:**
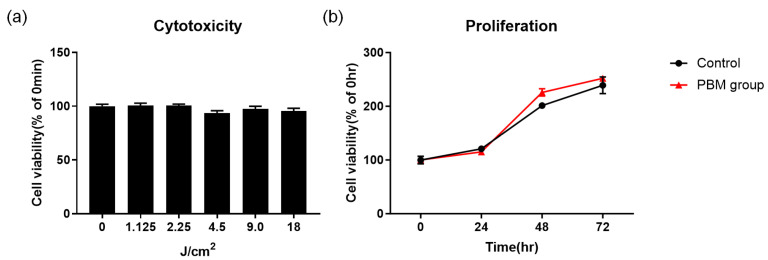
Multi-wavelength PBM has no toxicity on cell viability and cell proliferation of ARPE-19 cells. (**a**) Cell viability was measured after the irradiation of multi-wavelength PBM at an intensity of 0, 1.125, 2.25, 4.5, 9.0, and 18 J/cm^2^. (**b**) After the irradiation of multi-wavelength PBM at 4.5 J/cm^2^, cell proliferation was measured in control and PBM-treated groups for 24, 48 and 72 h.

**Figure 2 ijms-24-17394-f002:**
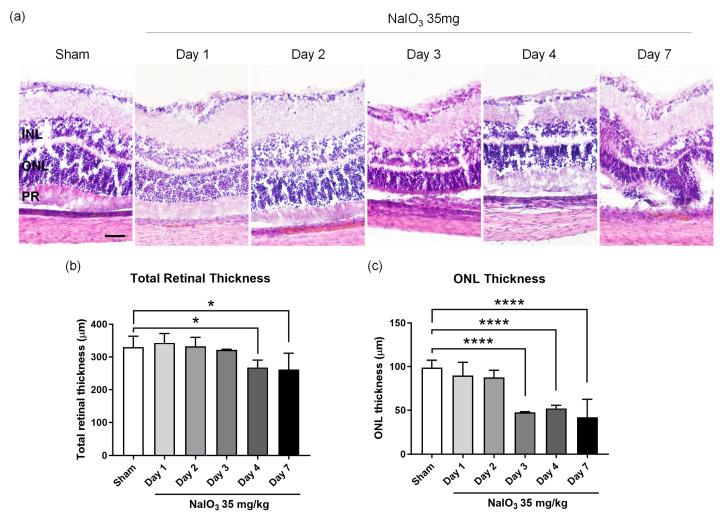
The histological analysis establishes the macular degeneration animal model following 7 days after NaIO_3_ injection. (**a**) The H&E-stained histological images showed a disruption in the ONL and photoreceptor layer in the retina. The scale bar indicates 50 µm. (**b**) The bar graph shows the total retinal thickness until 7 days after NaIO_3_ injection. (**c**) The bar graph shows the ONL thickness 7 days after NaIO_3_ injection. We established the macular degeneration animal model following 7 days after NaIO_3_ treatment, considering the total retinal thickness, ONL thickness, and disruption to the photoreceptor layer. INL, inner nuclear layer; ONL, outer nuclear layer; PR, photoreceptor layer. Data are expressed as the mean ± SEM, * *p* < 0.05, **** *p* < 0.0001, relative to sham treatment (ANOVA with the Bonferroni test).

**Figure 3 ijms-24-17394-f003:**
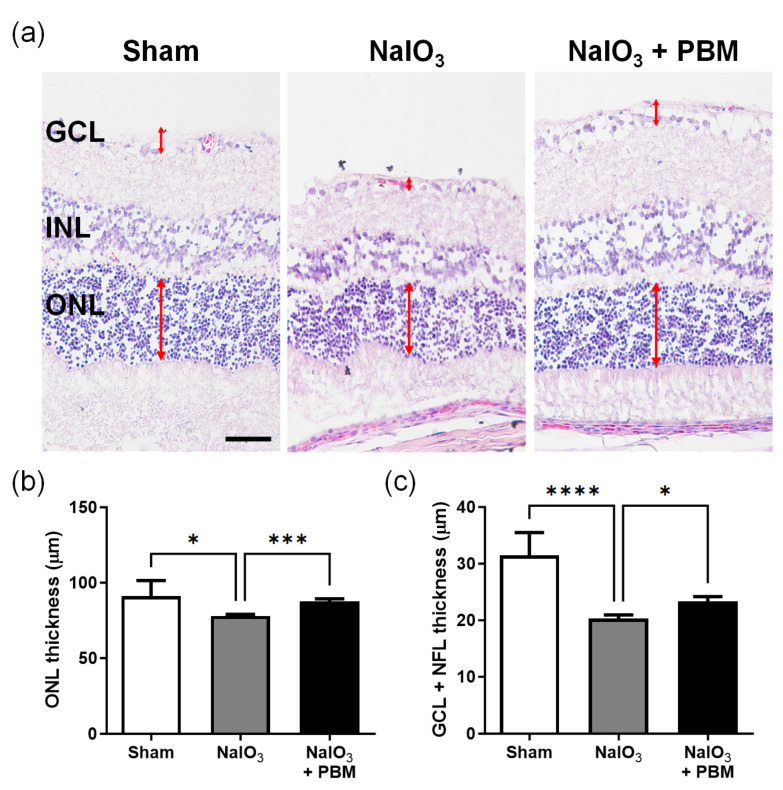
Effects of multi-wavelength LED treatment on the retina in NaIO_3_-induced macular degeneration. (**a**) Representative images of H&E-stained rat retina showed that multi-wavelength PBM protected the retina thickness against NaIO_3_-induced macular degeneration (red double-headed arrow; ONL and GCL + NFL thickness). The scale bar indicates 50 µm. (**b**) The bar graph shows significant changes in ONL thickness following irradiation with multi-wavelength PBM. (**c**) Quantitative analysis revealed that multi-wavelength PBM significantly increased the GCL and NFL thickness. GCL, ganglion cell layer; INL, inner nuclear layer; ONL, outer nuclear layer. Data are expressed as the mean ± SEM, * *p* < 0.05, *** *p* < 0.001, **** *p* < 0.0001, relative to sham treatment (ANOVA with the Bonferroni test).

**Figure 4 ijms-24-17394-f004:**
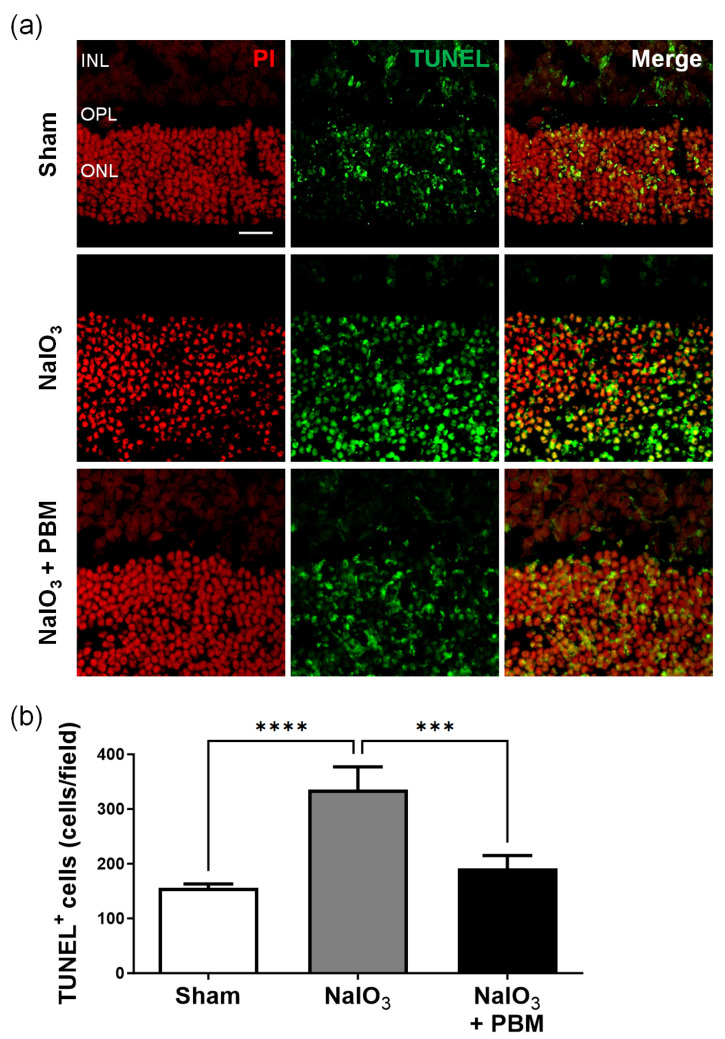
Multi-wavelength PBM reduced retinal apoptosis in ONL after NaIO_3_-induced macular degeneration. (**a**) Representative images of TUNEL (green) stained retina sections after NaIO_3_-induced macular degeneration. Irradiation with multi-wavelength PBM protected retinal cells of the ONL after NaIO_3_ injection. The scale bar indicates 20 µm. (**b**) Quantitative analysis of the apoptotic cells in ONL in the Sham, NaIO_3_, and NaIO_3_ + PBM groups. INL, inner nuclear layer; OPL, outer plexiform layer; ONL, outer nuclear layer. Data are expressed as the mean ± SEM, *** *p* < 0.001, **** *p* < 0.0001 relative to sham treatment (ANOVA with the Bonferroni test).

**Figure 5 ijms-24-17394-f005:**
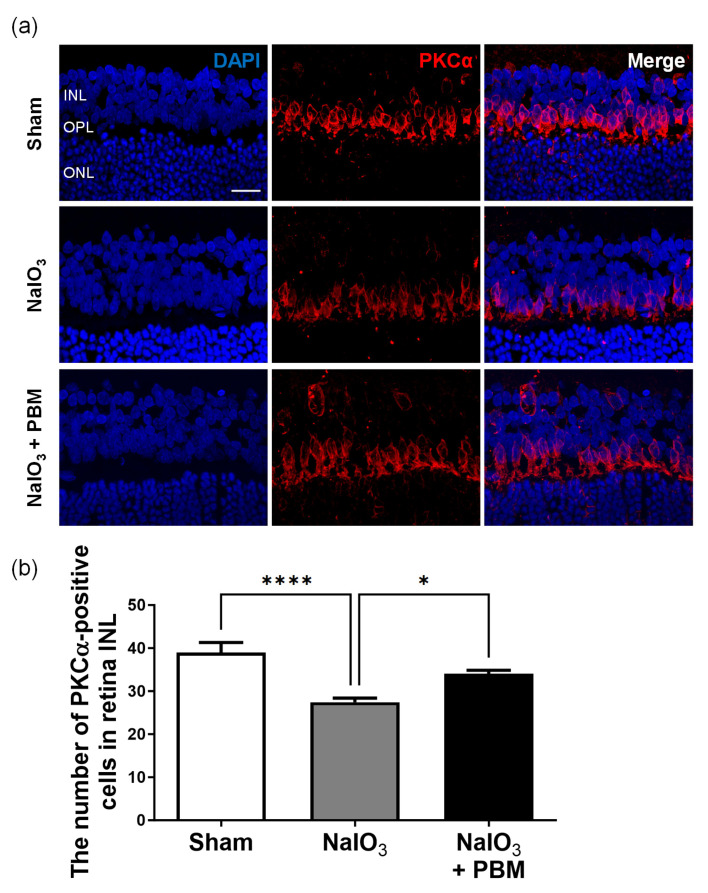
Multi-wavelength PBM attenuated the depletion of rod bipolar cells in retinal INL against NaIO_3_-induced macular degeneration. (**a**) Representative images of immunosignal PKCα (red) showed the rod bipolar cells in INL of the retina. The scale bar indicates 20 µm. (**b**) The number of PKCα-positive rod bipolar cells was quantified. This quantification revealed that the multi-wavelength PBM significantly increased the number of rod bipolar cells in the INL. INL, inner nuclear layer; OPL, outer plexiform layer; ONL, outer nuclear layer. Data are expressed as the mean ± SEM, * *p* < 0.05, **** *p* < 0.0001, relative to sham treatment (ANOVA with the Bonferroni test).

**Figure 6 ijms-24-17394-f006:**
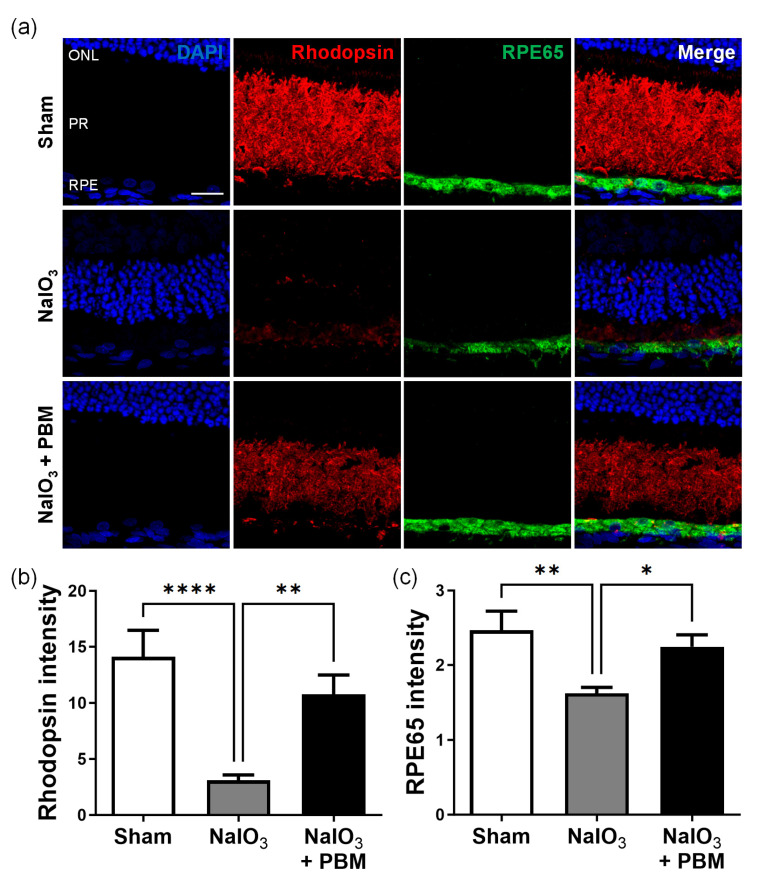
The inhibition of the photoreceptor degeneration by multi-wavelength PBM reduces the RPE toxicity in NaIO_3_-induced macular degeneration. (**a**) Representative immunofluorescent images for rhodopsin (red) and RPE65 (green) after NaIO_3_ injection and the irradiation of multi-wavelength PBM. The scale bar indicates 20 µm. (**b**,**c**) The bar graphs show the quantitative intensity analysis of rhodopsin and RPE65 after NaIO_3_ injection and the irradiation of multi-wavelength PBM. ONL, outer nuclear layer; PR, photoreceptor layer; RPE, retinal pigment epithelium. Data are expressed as the mean ± SEM, * *p* < 0.05, ** *p* < 0.01, **** *p* < 0.0001, relative to sham treatment (ANOVA with the Bonferroni test).

**Figure 7 ijms-24-17394-f007:**
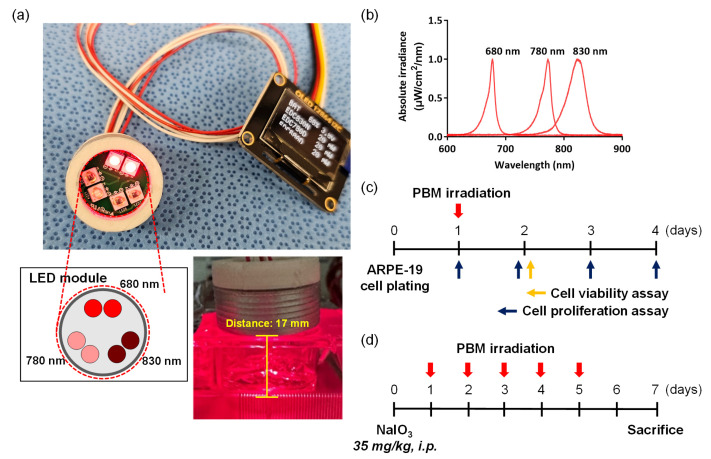
Multi-wavelength PBM irradiance and the experimental schedule for in vivo and in vitro study. (**a**) Experimental setup for treatment with multi-wavelength PBM. Two LED modules at wavelengths of 680, 780, and 830 nm are arranged in a single cylindrical irradiation chamber. (**b**) Beam profiling of multi-wavelength PBM using a spectrophotometer. The 680, 780, and 830 nm LED wavelengths were used to irradiate the ARPE-19 cells and NaIO_3_-induced macular degeneration in rats. (**c**) ARPE-19 cells were only irradiated at 1 day after cell plating. The 680, 780, and 830 nm LED wavelengths were irradiated at 17 mm from the bottom of the cell plate. (**d**) Multi-wavelength PBM was irradiated at 1 day after NaIO_3_ injection until day 5, and animals were sacrificed on day 7. Each cylindrical irradiation chamber was simultaneously applied to each eye of the rats.

## Data Availability

The data that support the findings of this study are available from the corresponding author upon reasonable request.

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
