# Peer review of "Multi-Wavelength Photobiomodulation Ameliorates Sodium Iodate-Induced Age-Related Macular Degeneration in Rats"

_ijms, 2023, doi:10.3390/ijms242417394_

Round 1

Reviewer 1 Report

Comments and Suggestions for Authors

Major comments

A graphical abstract will be very suitable for this manuscript.

Please insert a photo regarding to animal model of light exposure.

Please extend the discussion part

The animal model is not well presented in the text. Please exclude.

Why did you choose six weeks old rats?

Minor comments

Title should be changed as ‘…. Degeneration model in rats’

The following sentence should be excluded from the abstract ‘In vitro experiments confirmed that multi-wavelength PBM was not toxic to retinal pigment epithelial cells.

Page 3, Line 91, check spelling of NaIO3, same for the following ones within the text and figures.

Some upper and lower cases must be corrected. Please check within the text.

Reviewer 2 Report

Comments and Suggestions for Authors

1. Introduction Clarifications: - Prevalence and population bias: It would be beneficial to have some information on whether AMD affects specific populations disproportionately. For instance, are certain ethnicities or regions more prone to AMD? - Genetic factors: The role of genetics in AMD should be elaborated on. If there are known genetic markers or mutations that are associated with an increased risk of AMD, this information could be valuable in understanding the disease's etiology and in designing prevention strategies. - Preventative methods: The importance of prevention cannot be overstated. Understanding the current preventative methods, their efficacy, and any challenges associated with them will give context to the necessity of new therapeutic strategies like the PBM method being discussed.

2. Existing Therapies and Translation to Clinical Use: - Current PBM therapies: Have there been any other studies or clinical trials using PBM for AMD or other retinal diseases? What were their outcomes and how does this study compare? - Translation to therapy: PBM's clinical translation is crucial. It's important to understand if the authors foresee this treatment being a stand-alone therapy or if it would be combined with current therapies. Also, understanding the feasibility of delivering this treatment (e.g., would it require specialized equipment, trained personnel, etc.) would be beneficial. - Safety and efficacy: Long-term studies are necessary to understand any potential side effects or long-term benefits of the therapy. Do the authors anticipate any challenges in scaling their results from a rat model to human trials?

3. Current Treatments for AMD: - Dry vs. Wet AMD treatments: While the abstract mentions treatments primarily for wet AMD, the authors should provide a brief overview of the most common treatments for both types. - Efficacy of current treatments: Understanding the success rate and limitations of current treatments will provide context. For example, how often do current treatments restore vision versus merely halting or slowing the progression of the disease? - Side effects and limitations: Every treatment comes with its set of challenges. Whether it's potential side effects, cost, accessibility, or the need for repeated treatments, understanding the challenges of current treatments can emphasize the importance of novel strategies like PBM. - Comparison with PBM: How does PBM stack up against these treatments in terms of efficacy, potential side effects, and feasibility?

4. Mechanism of Action: The authors could delve deeper into how PBM acts at a cellular level. While the anti-inflammatory and antioxidant effects are mentioned, a more in-depth exploration of the underlying cellular and molecular mechanisms might provide a clearer picture of its therapeutic potential.

5. Limitations of the Current Study: It's essential for the authors to address the limitations of their study explicitly. For instance, are there any concerns about the rat model's applicability to human AMD? Are there potential confounding factors they haven't addressed?

6.     Dose-Response Relationship: Did the authors test various intensities and durations of PBM? Often, the effects of treatments are dose-dependent, meaning that while a certain intensity or duration might not be cytotoxic, higher or prolonged exposures might be. Understanding this relationship would give more confidence in the safety of PBM.

7.     Cell Line vs. Primary Cells: Was the in vitro experiment conducted using an immortalized RPE cell line or primary RPE cells? Cell lines sometimes behave differently than primary cells, which could impact the generalizability of the results.

8.     Short-Term vs. Long-Term Effects: Lack of immediate cytotoxicity doesn't necessarily mean there won't be long-term effects on cell function or viability. Were the cells monitored over an extended period post-PBM exposure to ensure no delayed effects or subtle cellular dysfunctions?

9.     Cellular Functions: Apart from cytotoxicity, it's essential to assess if PBM affects other cellular functions. For example, did the treatment alter the phagocytic function of RPE cells, their secretion profile, or their ability to support photoreceptors?

10.  Duration of Analysis: The 1 to 7 days duration for analysis might capture acute changes post-NaIO3 treatment, but longer-term studies would provide insights into chronic changes and potential recovery, this should be mentioned by authors and and thought about in future directions

11.  Future Directions: Following this study, what are the next steps? Would the authors consider testing different PBM wavelengths, durations, or intensities? Do they anticipate moving to primate models or early-stage human trials?

Round 2

Reviewer 1 Report

Comments and Suggestions for Authors

Dear Authors,

Thank you very much for your great effort.